# Estimating Sleep Stages using a Head Acceleration Sensor

**DOI:** 10.3390/s21030952

**Published:** 2021-02-01

**Authors:** Motoki Yoshihi, Shima Okada, Tianyi Wang, Toshihiro Kitajima, Masaaki Makikawa

**Affiliations:** 1Department of Robotics, Faculty of Science and Engineering, Ritsumeikan University Graduate Schools, Shiga 525-8577, Japan; 2Department of Robotics, Faculty of Science and Engineering, Ritsumeikan University, Shiga 525-8577, Japan; s-okada@fc.ritsumei.ac.jp (S.O.); t-wang@fc.ritsumei.ac.jp (T.W.); makikawa@se.ritsumei.ac.jp (M.M.); 3MD-4 Lab., Samsung R&D Institute Japan, Osaka 562-0036, Japan; t.kitajima@samsung.com

**Keywords:** ballistocardiogram, head acceleration sensor, sleep stages, sleep disruption, REM sleep

## Abstract

Sleep disruption from causes, such as changes in lifestyle, stress from aging, family issues, or life pressures are a growing phenomenon that can lead to serious health problems. As such, sleep disorders need to be identified and addressed early on. In recent years, studies have investigated sleep patterns through body movement information collected by wristwatch-type devices or cameras. However, these methods capture only the individual’s awake and sleep states and lack sufficient information to identify specific sleep stages. The aim of this study was to use a 3-axis accelerometer attached to an individual’s head to capture information that can identify three specific sleep stages: rapid eye movement (REM) sleep, light sleep, and deep sleep. These stages are measured by heart rate features captured by a ballistocardiogram and body movement. The sleep experiment was conducted for two nights among eight healthy adult men. According to the leave-one-out cross-validation results, the F-scores were: awake 76.6%, REM sleep 52.7%, light sleep 78.2%, and deep sleep 67.8%. The accuracy was 74.6% for the four estimates. This proposed measurement system was able to estimate the sleep stages with high accuracy simply by using the acceleration in the individual’s head.

## 1. Introduction

Sleep accounts for about one-third of human life. Sleep disruptions from causes such as changes in lifestyle, stress from aging, family issues, or life pressures, is a growing phenomenon that can lead to serious health problems. Insomnia, one of the sleep disorders, affects about 33% of the general population [1]. Sleep disorders can lead to serious health issues including an increased risk of death [2]. As such, sleep disorders need to be identified and addressed early on.

The disruption of sleep rhythms can be determined by measuring sleep stages [3]. In clinical practice, multiple electrodes are attached to the individual and biological signals are measured to determine sleep stages. This approach requires a technician with specialized knowledge and experience, and the sleep stage determination test is laborious and complicated. As the individual needs to go to bed with electrodes and an electroencephalograph attached, the examination itself may affect normal sleep patterns as well. Therefore, sleep patterns have recently been investigated by using a wristwatch-type device or a camera to capture body movement information. However, this method only discriminates between the individual’s asleep and awake states [4,5]. Other researchers have found that by using time-series data on gross movements, such as turning over during sleep, it is possible to estimate other sleep stages. However, this is based on the knowledge that gross movements tend to increase a few minutes before the start of the rapid eye movement (REM) sleep stage and the longer the appearance interval of the gross movements, the deeper the non-REM sleep. Hence, it is difficult to achieve a high level of accuracy with this approach as the information is based only on the tendencies of the sleep stages [6].

To improve on this, we focus on the movement of the twitch of the head. This twitch movement is part of the ballistocardiogram (BCG) that captures the body’s mechanical response to the physical movements of the heart and the pumping of blood. The J wave, which is the prominent part of the BCG, represents the acceleration of blood in the descending aorta and abdominal aorta and the deceleration of blood in the ascending aorta [7]. An individual’s heart rate interval can be derived from the J-J interval (JJI) [8]. Studies have confirmed that one’s heart rate interval fluctuates dramatically, even during sleep, due to the influence of the autonomic nervous system [9]. As this relationship with sleep stages has been shown, as well, the heart rate interval can be a factor that can be used to effectively estimate other stages of sleep beyond the distinction between being asleep and being awake. The human body also has resonance frequencies, as muscles and bones act as springs and dampers [10]. During REM sleep, all one’s muscles, such as eyes and respiratory muscles, are relaxed, so the human body’s vibrations are transmitted in minute movements such as breathing and BCG changes [3]. Our supposition is that REM sleep can be captured by analyzing the frequency of these characteristics. There are three reasons why we focused on the head among the twitch movements. Firstly, according to Kato et al., the rate of gross movements in overall sleep is 39.39% for the limbs, 27.43% for the trunk, and 22.91% for the head [11]. Therefore, when measuring the pulsation with the arm, it is possible that body movement noise due to the movement of the limbs will be included and the pulsation measurement system will be lowered. Moreover, since the head has fewer large movements than the arms, the influence of pulsation measurement is small. Secondly, a study by Yousefian et al. found that wrist BCG was affected by arm mass, spinal damping, and arm stiffness [12]. In the lateral decubitus position, the trunk may fix the arm and make BCG measurement difficult. The movement of the head by BCG is a vertical movement [13], and it is not easily affected by the posture during sleep. Therefore, the measurement of BCG during sleep is better on the head than on the wrist. Finally, the twitch movements of the head can be measured by using an earphone-type accelerometer [8] or camera [13,14], which is practical.

Our study objective is to estimate sleep stages by using two features, body movements and heart rate information, derived only from the acceleration in an individual’s head. The heart rate information is collected from twitch movements, including a BCG of the individual’s head. In recent years, there is a wristwatch-type device of Fitbit Inc. for easy monitoring of daily sleep, but they are a little different from the polysomnography (PSG) test data used clinically [15]. Therefore, in this study, we conducted a comparative experiment with the sleep stages used in the PSG test, which is also used clinically.

The rest of the paper is organized as follows. Section 2 discusses our methods and subjects, how we created correct answer data for supervised learning, how we extracted features, how we created a classifier, and how we performed the evaluation. Section 3 presents the results and Section 4 our discussion. Section 5 concludes.

## 2. Subjects and Methodology

### 2.1. Subjects 

Sleep is affected by age [16,17], gender [18], marriage [19], and working stress [20]. Therefore, we recruited unmarried experimental subjects, who live a regular life and are physically and mentally healthy, with posters at Ritsumeikan University. Eight healthy male students (age 20–23 years) participated in the experiment. Table 1 shows the subject information. Each subject underwent three nighttime sleep experiments. Since sleeping in an unfamiliar place may change sleep patterns and REM sleep [3], the first night was excluded and only the records for the second and third nights were used for our analysis. The subjects were given sufficient informed consent in advance to ensure there were no sleep-inhibiting factors such as pre-experimental caffeine intake and excessive exercise. The study was conducted with the approval of the BKC Research Ethics Review Committee (BKC-2019-076).

### 2.2. Experiment

Subjects were instructed to go to bed by 23:00 for one week before the experiment. Furthermore, subjects were restricted from caffeine intake, drinking, and excessive exercise for 3 h before the experiment. In addition, each subject decided the experiment date to reduce their stress. As shown in Figure 1, the experimental environment was a private room where the temperature was 22–24 °C, the humidity 50–60%, the illuminance was 3 lux or less, and the noise was 40 dB or less, the conditions used in a general sleep stage test [21]. During the sleep experiment, only the study subjects were allowed in the room. 

The experimental equipment used were a biological amplifier Polymate Pro 6000 (Digitex Lab. Co., Ltd., Tokyo, Japan) and the KXM52-1050, 3-axis accelerometer module (Kionix, Tokyo, Japan measurement range ±19.6 m/s^2^, sensitivity 0.66 V/(m/s^2^)). Since the acceleration of the head by BCG is in the range of 0.98 m/s^2^ [22], KXM52-1050 was selected. An electroencephalogram (EEG) based on the international 10–20 method (C3-A2, C4-A1, O1-A2, O2-A1), an electrooculogram (EOG), an electromyogram (EMG) of the chin, and an electrocardiogram (ECG) were measured using the bio-amplifier of Polymate Pro 6000. The ECG was not used to create an automatic sleep classifier but rather for comparison with the JJI, which could be gathered from the BCG. As an external input of the Polymate Pro 6000, 3-axis acceleration was recorded at 200 Hz. The 3-axis accelerometer was directly fixed to the center of the forehead using biological tape and had little effect on bedtime. For safety, the sponge and the sensor were bonded with silicon.

### 2.3. Methodology 

#### 2.3.1. Defining the Sleep Stage

When estimating sleep stages, sleep stage data are required as teaching data. For the sleep stages in this study, the EEG, EOG, and EMG of the chin were collected as one epoch each of 30 s. Sleep stages were determined using the American Academy of Sleep Medicine (AASM) Manual for the Scoring of Sleep and Associated Events [23]. As shown in Figure 2, we used four sleep stages, with WAKE for the awake stage, REM for the REM sleep stage, LIGHT for the non-REM sleep stage N1-2, and DEEP for the non-REM sleep stage N3. REM, LIGHT, and DEEP, which are all sleep states, are collectively referred to as SLEEP.

#### 2.3.2. Sleep Stage Estimating

We used a two-step classification method in this experiment. In this two-step method, first, classifier 1 is used to estimate the two states: WAKE and SLEEP. Next, in classifier 2, the three states of REM, LIGHT, and DEEP are estimated in SLEEP using the features extracted from the results of classifier 1. Finally, sleep is classified into four states from the two steps (Figure 3). We applied the two-step classification method because we needed to extract features based on the first results of WAKE and SLEEP. In addition, the difference between the WAKE and SLEEP states was considerably larger than the differences among REM, LIGHT, and DEEP sleep states [3]; therefore, we could improve our estimation accuracy if WAKE and SLEEP were classified first.

There were five features used in the first-step to classify WAKE and SLEEP as shown in Table 2. 

Two of these were extracted from gross movements (Nos. 1 and 2), which included body movement information, such as turning over; one was extracted from frequency (No. 3); and two were extracted from twitch movements, including the BCG (Nos. 4 and 5).

For the features extracted from gross movements, a 0.1 Hz high-pass filter was first used for the 3-axis acceleration raw data. The purpose of this was to remove noise and direct current (DC) components from the EMG. Next, the root mean square (RMS) was used for the 3-axis acceleration after bandpass filtering, with the acceleration converted to 1-axis. This reduces the effects of peak emphasis, changes in acceleration due to sleep posture, and changes in acceleration due to errors during sensor installation. Finally, to unify the sampling time with the sleep stage data, the mean (average of gross movements: AGM) and variance (variance of gross movements: VGM) were calculated every 30 s. For the features extracted from frequency, the discrete Fourier transform was performed every 30 s on the data centered on one axis by the RMS to obtain the total spectrum of all frequency bands (Full spectrum: FS). These data are considered effective in estimating the WAKE stage as they capture the tendency to wake up when gross movements are frequent [5].

The features that could be extracted from twitch movements were those from the JJI. To do this, we obtained the JJI following the procedure shown in Figure 4. In preparing for peak detection, a 1–10 Hz bandpass filter was used first for the 3-axis acceleration raw data. As in the case of gross movements, this is to remove noise, DC components, and respiratory components from the EMG. Next, the RMS is used for 3-axis acceleration after bandpass filtering and converted to 1-axis. This again reduces the effects of peak emphasis, changes in acceleration due to sleep posture, and changes in acceleration due to errors during sensor installation. Finally, we used a moving average filter with a window width of 0.325 s for the RMS data. This smoothed data to prevent over-detection during peak detection.

In the peak detection process, the heartbeat has a refractory period of approximately 0.2 s [7], with the heartbeat interval more stable during sleep than during exercise. Therefore, maximum value detection was performed with a minimum peak detection interval of 0.365 s. A median filter every 0.365 s was used as the peak correction. We captured the J wave, which was the peak value, and the JJI.

Because the JJI does not occur at equal time intervals, it was resampled to 2 Hz using cubic spline interpolation. When extracting the features that will be effective for estimating sleep stages from heart rate intervals, frequency analysis can be performed for each heart rate interval epoch [9]. This means that a sampling frequency above a certain level is required. However, as there was a delay of 0.1–0.3 s between the heartbeat interval obtained from the BCG and the heartbeat interval obtained from the ECG [8], resampling was set to 2 Hz, to avoid the error that would occur if the sampling frequency was higher than necessary. Finally, a median filter with a window width of 7 s was used for outlier processing.

After calculating the JJI, its average (Average of JJI: AJJI) and its variance (Variance of JJI: VJJI) every 30 s were calculated to unify the sampling time of the sleep stage data and the JJI data. For the features extracted from the twitch movements, when gross movements occur, these twitch movements become difficult to detect and an abnormal value appears. Therefore, when gross movements appear frequently, the tendency to wake up [6] can be detected by abnormal values of body movements.

The features for the WAKE and SLEEP estimations were all parameters that used gross movements, including turning over.

There were nine features in the second-step classifier, which classify REM, LIGHT, and DEEP sleep, as shown in Table 3. Two features could be extracted from gross movements (Table 3, Nos. 1–2), which include body movement information, such as turning over; one could be extracted from frequency (Table 3, No. 3); four could be extracted from twitch movements, including the BCG (Table 3, Nos. 4–7); and two from the results in the first-step classifier (Table 3, Nos. 8–9).

We captured the features in Table 3 as follows. Nos. 1–5 were obtained by standardizing the features in Table 2 with an average of 0 and a variance of 1. We did not standardize the features of the first-step classifier for two reasons. The first is that the acceleration in the head is larger in gross movements than in twitch movements; therefore, if the WAKE data, including most of the gross movements, are not removed, they become a standardization that depends on the amount of WAKE data. Second, since there is little difference in the features among REM, LIGHT, and DEEP sleep, we needed to standardize and reduce errors between subjects. Moreover, as WAKE is clearly different from the other sleep states, there is no need to standardize it to reduce individual differences. For Nos. 6 and 7, after calculating the JJI as shown in Figure 4, we performed a discrete Fourier transform every 30 s to obtain a high frequency (HF, 0.15–0.4 Hz band spectrum), low frequency (LF, 0.04–0.15 Hz band spectrum), and total frequency (TF, 0.04–0.4 Hz band spectrum), used then to calculate the HF/TF and HF/LF. No. 8, a feature obtained from the first estimate, is the elapsed time since the first estimate of sleep. No. 9, also obtained from the first estimation, is calculated based on the duration of sleep.

The features shown in Table 3 are effective for estimating the following sleep stages. The SAGM (Table 3, No. 1) and the SVGM (Table 3, No. 2) are considered effective for estimating DEEP sleep. These features capture that DEEP sleep has less head movement than REM and LIGHT sleep [6]. The SFS (Table 3, No. 3) is considered effective for estimating REM, LIGHT, and DEEP sleep. Since the BCG of the head is considered proportional to the force of blood pumped by the heart [14], the supposition is that the spectrum of each sleep stage will change according to the spectrum of the acceleration of the head. The SAJJI (Table 3, No. 4) is considered effective for estimating REM, LIGHT, and DEEP sleep. Since SAJJI is the average of the JJI every 30 s, it contains information close to very low frequency (VLF: spectrum of the frequency band of 0.0033–0.04 Hz) of the heartbeat interval. Studies have shown that VLF is associated with blood pressure regulation [24] and is influenced not only by the autonomic nervous system activity but also by random physical activity [25]. Our aim was to capture its characteristics via the SAJJI. We did not obtain the VLF through spectral analysis from the JJI because the sampling frequency was too low. The SVJJI (Table 3, No. 5) is considered effective for estimating REM sleep. During REM sleep, the autonomic nervous system and the heartbeat interval are disturbed [3]. The SHF/TF (Table 3, No. 6) and SHF/LF (Table 3, No. 7) are considered effective for estimating REM, LIGHT, and DEEP sleep. The HF/TF is commonly used as an indicator of the parasympathetic nervous system and increases from REM to DEEP sleep [26]. HF/LF is commonly used as an indicator of the sympathetic nervous system and decreases from REM to DEEP [25]. SET (Table 3, No. 8) is considered effective for estimating REM, LIGHT, and DEEP sleep. DEEP sleep increases in the first half of sleep and decreases in the second half while REM sleep increases [3]. HRT (Table 3, No 9) is considered effective for estimating DEEP sleep. This is because there are changes, such as a decrease in body movements, about 10 min before DEEP sleep. These body movements decrease because DEEP sleep is deeper than REM and LIGHT sleep [6].

The classifier used a random forest technique (module uses scikit-learn), an ensemble learning method, as it is easy to check the contribution rate of the features, and how each feature affects each sleep stage. In addition, the effect of overfitting is reduced compared with a decision tree, and slight fluctuations in features have less effect on the classifier. Deep learning was not used as there were little training data. The random forest hyperparameters were adjusted through a grid search. In the grid search, we performed leave-one-out cross-validation using data from all subjects and obtained the hyperparameters that maximize the F-score. There were three hyperparameters that needed to be adjusted: threshold determination method (Gini coefficient, entropy), tree depth (1–10), and number of trees (1–10). The other hyperparameters were the initial values of the scikit-learn random forest.

#### 2.3.3. Evaluation Methods

To evaluate the features in the first-step classifier, features every 20 epochs were randomly extracted from each sleep stage and a Wilcoxon-signed rank test was performed. For the evaluation in the second-step classifier, features every 20 epochs were randomly extracted from each sleep stage and the Freidman test was performed. If a significant difference was found, multiple comparisons were made using the Wilcoxon signed-rank test and Bonferroni correction was performed. The significance level was <5%.

An evaluation of the classifier was created using leave-one-out cross-validation to obtain the F-score, precision, and recall of each sleep state and their accuracy. In addition, leave-one-out cross-validation was performed on all subjects to show the overall performance of the classifier, and the test results of each iteration were averaged to obtain receiver operation characteristic (ROC) curves and areas under curves (AUCs) were calculated. Leave-one-out cross-validation was done using data (*n* = 14) of seven subjects for two nights as training data and data of the one remaining subject for two nights as test data. Leave-one-out cross-validation is more practical than k-cross-validation because the test and training data are from completely different subjects. However, if the training dataset is small, accuracy will be low because it will depend on individual differences among the subjects. In addition, we chose leave-one-out cross-validation since k-fold cross-validation could not be used due to our two-step classification.

## 3. Results

### 3.1. Results of Sleep Stages Used as Correct Data

Table 4 shows the number of sleep stage epochs on the second and third nights for each subject. We can see that there is not a large difference in the ratio of sleep stages between the second night and the third night. Only on the third night for subject 3, REM is extremely short compared with the general time [3], and it is clear that this is not normal sleep. Therefore, the subsequent analysis excludes the third night of subject 3. 

### 3.2. Relationship between Features of First-Step Classifier and Sleep Stages

Figure 5 shows the measured data of the BCG and ECG during sleep. The BCG captures the data obtained through steps 1–5 in Figure 4. The red points on the BCG reflect the acquired J wave, and the red points on the ECG reflect the acquired R wave. As shown in Figure 5a, there is little difference in the JJI and RR intervals (RRI) between the BCG and the ECG in the resting state. However, as shown in Figure 5b, there are few J waves acquired in the BCG, including gross movements.

Figure 6 shows the relationship between the features of the first-step classifier on the second night for subject 1 and his sleep stages. As shown in Figure 6a–c,e, the estimated WAKE values tended to be considerably larger than the estimated SLEEP values. In contrast, for the AJJI and other features shown in Figure 6d, there tended to be a smaller difference in values between WAKE and SLEEP. The AJJI tended to have many abnormal values in the WAKE estimation. In addition, it showed that the value normally acquired during WAKE decreased.

Figure 7 shows the results of the Wilcoxon signed rank test for each feature in the first-step classifier. There were highly significant differences in all the features: AVG (1.88 × 10^−1^ ± 0.20 × 10^−1^ m/s^2^ and 1.87 × 10^−2^ ± 7.37 × 10^−3^ m/s^2^, *p* < 0.01), VGM (1.12 × 10^−1^ ± 1.94 × 10^−1^ (m/s^2^)^2^ and 3.24 × 10^−4^ ± 2.07 × 10^−3^ (m/s^2^)^2^, *p* < 0.01), FS (1.83 ± 1.56 (m/s^2^)/Hz and 1.85 × 10^−1^ ± 9.40 × 10^−2^ (m/s^2^)/Hz, *p* < 0.01),AJJI (1.22 ± 1.72 × 10^−1^ s and 1.12 ± 1.80 × 10^−1^ s, *p* < 0.01), and VJI (2.19 × 10^−1^ ± 1.93 × 10^−1^ s^2^ and 1.70 × 10^−2^ ± 2.80 × 10^−2^ s^2^, *p* < 0.01).

### 3.3. Relationship between Features of Second-Step Classifier and Sleep Stages

Figure 8 shows the relationship between the features of the second-step classifier on the second night for subject 1 and the sleep stages. Although, the second-step classifier includes only data estimated as SLEEP in the first-step classifier, Figure 8 also includes WAKE data. Figure 9 shows the results of the Wilcoxon signed-rank test for each feature of the second-step classifier.

Figure 8a–c reflect large values before and after WAKE. However, as shown in Figure 9a–c, the REM sleep of the SAGM was −5.39 × 10^−2^ ± 5.02 × 10^−1^, the LIGHT sleep of the SAGM was −4.68 × 10^−2^ ± 7.76 × 10^−1^, and the DEEP sleep of SAGM was 4.18 × 10^−2^ ± 1.48. The REM sleep of the SVGM was −7.30 × 10^−2^ ± 1.97 × 10^−1^, the LIGHT sleep of the SVGM was 6.25 × 10^−4^ ± 6.65 × 10^−1^ and the DEEP sleep of the SVGM was 9.96 × 10^2^ ± 1.81. The REM sleep of the SFS was −7.68 × 10^−2^ ± 5.5 × 10^−1^, the LIGHT sleep of the SFS was −1.96 × 10^−2^ ± 9.03 × 10^−1^, and the DEEP sleep of the SFS was 1.70 × 10^−2^ ± 1.46; as *p* > 0.05 among REM, LIGHT, and DEEP sleep stages, there was no significant difference.

As shown in Figure 8d, the SAJJI value increases as it approaches DEEP sleep and decreases as it approaches REM sleep. In Figure 9d, the REM sleep of SAJJI was −5.92 × 10^−1^ ± 1.18, the LIGHT sleep of the SAJJI was 1.48 × 10^−1^ ± 9.45 × 10^−1^, and the DEEP sleep of the SAJJI was 1.86 × 10^−2^ ± 7.92 × 10^−1^. Except for LIGHT sleep and DEEP sleep, *p* < 0.01; thus, it was highly significant.

As shown in Figure 8e, the SVJJI value tended to be stable as it approached DEEP sleep, and the value tended to diverge as it approached WAKE and REM sleep. In Figure 9e, the REM sleep of the SVJJI was 1.13 × 10^−1^ ± 8.34 × 10^−1^, the LIGHT sleep of the SVJJI was −9.61 × 10^−2^ ± 6.85 × 10^−1^, and the DEEP sleep of the SVJJI was −4.27 × 10^−2^ ± 7.99 × 10^−1^. Except for LIGHT and DEEP sleep, *p* < 0.01; thus, it was highly significant. 

Figure 8f,g had similar shapes, with the values increasing as they approached DEEP sleep and decreasing as they approached REM sleep, but the signal-to-noise (S/N) ratio was low. In Figure 9f,g, the REM sleep of the SHF/TF was −2.66 × 10^−1^ ± 9.76 × 10^−1^, the LIGHT sleep of the SHF/TF was 1.78 × 10^−2^ ± 1.00, and the DEEP sleep of the SHF/TF was 2.54 × 10^−1^ ± 9.19 × 10^−1^. The REM sleep of the SHF/LF was −2.64 × 10^−1^ ± 7.88 × 10^−1^, the LIGHT sleep of the SHF/LF was 1.17 × 10^−2^ ± 9.68 × 10^−1^, and the DEEP sleep of the SHF/LF was 2.31 × 10^−1^ ± 1.06. Except for the LIGHT and DEEP sleep, *p* < 0.01; thus, it was highly significant. In addition, between the LIGHT and DEEP sleep stages, *p* < 0.05; thus, it was significant. 

Figure 8h was linear, as it reflects the elapsed sleep time. DEEP sleep increased as the SET value decreased, and REM sleep increased as the SET value increased. As shown in Figure 9h, the REM sleep of the ST was 2.57 × 10 ± 2.56 × 10 min, the LIGHT sleep of the ST was 2.39 × 10 ± 3.15 × 10 min, and the DEEP sleep of the ST was 4.24 × 10 ± 2.68 × 10 min. Among REM, LIGHT, and DEEP sleep stages, *p* < 0.01; thus, it was highly significant.

The HRT shown in Figure 8i has a lower value for REM and LIGHT sleep stages than for DEEP sleep. In Figure 9i, the REM sleep of the HRT was 2.87 × 10 ± 9.48 × 10 min, the LIGHT sleep was 2.22 × 10 ± 1.24 × 10 min, and the DEEP sleep was 1.49 × 10 ± 1.22 × 10 min. Except for the REM and LIGHT sleep stages, *p* < 0.01; thus, it was highly significant.

### 3.4. Results of Sleep Stage Estimations

Figure 10 shows the results of the comparison between the sleep stage estimations for subject 1 on the second night and the correct answer data. The basis was estimated as LIGHT sleep, which accounted for most of the sleep stage. WAKE had a high estimation for all measurements. REM sleep could be estimated intermittently, but not continuously. The estimation of DEEP sleep was high in the first half of the measurement, but it could not be estimated in the second half of the measurement.

Figure 11 shows the estimation results of all data for subject 3, except the third night, showing F-score, precision, and recall for each sleep stage and the accuracy. The average value of each result was 74.6% for Accuracy, 72.6% for all sleep stages F-score (AF), 75.1% for the WAKE F-score (WF), 76.6% for WAKE precision (WP), 72.6% for WAKE recall (WR), 52.7% for the REM F-score (RF), 61.3% for REM precision (RP), 49.0% for REM recall (RR), 78.2% for the LIGHT F-score (LF), 71.2% for LIGHT precision (LP), 87.4% for LIGHT recall (LR), 67.8% for the DEEP F-score (DF), 73.0% for DEEP precision (DP), and 65.6% for DEEP recall (DR).

Table 5 shows the results of the confusion matrix obtained from leave-one-out cross-validation for all data. Most of the mistakes from estimating WAKE, REM, and DEEP were estimated to be LIGHT. Most of the mistakes from estimating LIGHT were estimated to be REM and DEEP.

Figure 12 shows the contribution rate of the created classifier. In Figure 12a, VGM, FS, and AGM reflect 90% or more of the contribution of the WAKE and SLEEP estimates; the parameter using JJI had a contribution of 10% or less. In Figure 12b, the contribution of the REM, LIGHT, and DEEP sleep estimates for ET and ST, which were the features obtained from the estimation results in the first-step, were 50% or more. SAJJI contributed about 15% to the estimation, and the contribution of SVJJI, SVGM, SFS, and SFGM were less than 10%. The contribution of SHF/LF and SHF/TF were less than 5%, with little effect on the estimation. 

Figure 13 shows the results of the ROC curves. There is not much bias in the True positive rate and False positive rate. The AUCs were 0.98 for WAKE, 0.88 for REM, 0.81 for LIGHT, and 0.93 for DEEP.

## 4. Discussion

As shown in Table 4, the sleep stages of the subjects were all within the estimated ranges [3] except for the third night for subject 3. As shown in Section 2.2, subjects were given various restrictions to improve their consistency. From these, we can suppose that our data acquired through this experiment reflect general data for healthy subjects. Excluding the third night for subject 3, the number of datasets was 15, but in comparison with the studies of Cabon et al. [27] and Ran et al. [28], this is still sufficient to create an automatic sleep classifier under age- and gender-restricted conditions. However, to create an automatic sleep classifier that can handle various patterns, such as other genders, ages, and people suffering from sleep disorders, the number of subjects would need to be increased. Since most of the sleep states were LIGHT as shown in Table 4, they were often erroneously estimated as LIGHT as shown in Table 5. These results were similar to those of Supratak et al. [29].

From Figure 6, Figure 7, and Figure 12a, we see that in estimating WAKE and SLEEP stages, the features including gross movements, such as turning over, were more effective than changes in JJI and the error rate of JJI. This is because there is a tendency to wake up when gross movements are frequent [6]. In addition, as shown in the study by Alain et al. [30], the accuracy of the WAKE estimation is improved by using body movement information in addition to the heart rate interval. 

From Figure 8, Figure 9 and Figure 12b, we can see that it was effective to use the SET and HRT obtained from the WAKE and SLEEP estimations for REM, LIGHT, and DEEP sleep. This is because the sleep elapsed time is characterized by an increase in DEEP sleep in the first half of sleep time, a decrease in DEEP sleep in the second half, and an increase in REM sleep [3]. In addition, in the HRT, body movement decreases approximately 10 min before DEEP sleep as DEEP sleep is deeper than REM and LIGHT sleep [6].

Next, the SAJJI, as shown in Figure 8d and Figure 9d was an effective estimation. Since SAJJI was a feature that standardizes the average JJI every 30 s, it contains information close to the VLF of the heartbeat interval. The physiological meaning of VLF has not been clarified as much as the LF and HF, but Eckberg et al. [24] and Taylor et al. [31] point out that VLF acts as a renin-angiotensin system and as baroreceptor reflex sensitivity. Fleishere [32] published a paper suggesting that VLF reflects temperature regulation activity. According to the study by Johannes et al. [33], the VLF is significantly different among REM, LIGHT, and DEEP sleep, emphasizing its effectiveness for estimating these sleep stages. In addition, research has shown that sleep apnea can be detected by VLF [34], which is one of the merits of obtaining information close to VLF by using the acceleration in the head.

Figure 12b shows that SHF/TF and SHF/LF were not effective for estimating the REM, LIGHT, and DEEP sleep stages. There was also little difference between JJI and RRI in the BCG and ECG during the resting state, as shown in Figure 5a. Moreover, in Figure 5b, the J waves were hardly perceptible in the BCG, including gross movements. Therefore, when gross movements are included, the error of JJI becomes large, so it is necessary to correct it through averaging processing. In this way, although it can be used at low frequencies, it will be difficult to use at high frequencies because the detection accuracy for JJI is low. In recent years, research has investigated improving JJI detection accuracy using template matching [35]. With improved detection accuracy, estimation accuracy may increase if HF/TF and HF/LF features are available. 

As shown in Figure 9e, VJJI was significantly different in REM sleep than in the other sleep states. The reason is that during REM sleep, the autonomic nerves are characterized by irregular changes [3], which tend to result in irregular heartbeats and increased JJI dispersion. Moreover, as shown in Figure 13, this variance increased when gross movements were included, so other less affected features were more useful for estimating REM, LIGHT, and DEEP sleep stages.

As shown in Figure 8c, Figure 9c and Figure 12b, SFS was not effective for estimating REM, LIGHT, and DEEP sleep states. Moreover, there was no significant difference between the REM, LIGHT, and DEEP sleep states. As the BCG of the head is proportional to the force of blood pumped by the heart [14], the sequenced relationship for blood pressure and cerebral blood flow magnitude is REM > LIGHT > DEEP sleep [3]. Therefore, we assume that the spectral area of the BCG of the head comprises the relationship between REM > LIGHT > DEEP sleep. However, the classification became difficult due to errors between subjects and the measurement error due to the slight difference in these characteristics. For this reason, the SAGM and SVGM (Figure 8a,b, Figure 9a,b and Figure 12b) features using information on gross movements were not effective for estimating REM, LIGHT, and DEEP sleep, and there was no significant difference among the REM, LIGHT, and DEEP sleep states. 

The performance of our automatic sleep classifier reflected 74.6% accuracy, 72.6% all sleep stages F-score (Figure 11). Like our study, Willemen et al. [36], Zhang et al. [37], Nochino et al. [38], and Mitsukura et al. [35] have also developed automatic sleep classifiers that can estimate four states: WAKE, REM, LIGHT, and DEEP (Table 6). As a result, the accuracy of the study by Willemen et al. [36] was 69%, the all sleep stages F-score of the study by Zhang et al. [37] was 62%, and the study by Nochino et al. [38] was 41%. Therefore, our research had a better performance. However, using the BCG as we did for the sleep stage estimation, Mitsukura et al. had an accuracy of 89% [35]. In their research, they combined multiple high-performance sensors for their processing. In our study, we achieved our high accuracy sleep estimation by attaching only one inexpensive 3-axis accelerometer to the individual’s head. In addition, the WAKE F-score is 76.6% in this study and 70.2% in the study by Mitsukura et al. [35], and this study has a better performance. In the study by Nochino et al. [38], the performance of REM was poor because only body movement was used as a feature. Since there is little difference in body movement between REM and LIGHT [6], it is considered that the performance will improve if heart rate information is added as in this study.

As shown in Figure 10, continuous REM and DEEP sleep estimates are difficult. The reason is that the time resolution of the measured data were high, while the correct answer data, PSG, were a value once every 30 s. Therefore, if the correct answer data defined by PSG have the same resolution as the measurement data, it may match the estimation result. In addition, our belief is that performance can be improved by using bidirectional long short-term memory (BLSTM) used by Supratak et al. [29], which can use epochs before and after the estimated epoch. 

As shown in Figure 13, the ROC curve and the AUCs results prove that the proposed method is discriminative in the case of data unbalance.

However, the database for this measurement is different from that used in previous studies. Although our algorithms provide higher accuracy than traditional algorithms, this does not necessarily mean that they are superior; moreover, this high accuracy could be simply from the dataset.

## 5. Conclusions

In this study, an inexpensive 3-axis accelerometer was attached to the individual’s head, and the sleep stages were estimated by utilizing features of heart rate information derived from the BCG and body movement.

Based on the leave-one-out cross-validation, our results showed that the estimation accuracy was 74.6%, the WAKE F-score was 76.6%, the REM sleep F-score was 52.7%, the LIGHT sleep F-score was 78.2%, and the DEEP sleep F-score was 67.8%. Our supposition is that this estimating performance can be improved by using BLSTM. From these, if the device can acquire gross movements from the acceleration of the head, including turning over and twitch movements, along with the BCG, better study estimation accuracy will be possible.

Our system was a prototype and was wired to build a stable communication environment. In addition, from the viewpoint of ease of measurement, we conducted an experiment by fixing the sensor in the center of the forehead. From the viewpoint of ease of measurement, it is not practical because the sensor was fixed in the center of the forehead and the experiment was conducted. In the future, we are aiming to realize and put into practical use a similar system by measuring the acceleration of the head with an earphone equipped with a 3-axis accelerometer used by He et al. [8].

In addition, the system of this experiment is not general because various restraint conditions were set in order to improve the consistency of the subjects in this experiment. A wide variety of data are required to make it versatile.

Finally, there is a previous study showing that VLF of heart rate interval obtained from BCG can detect sleep apnea disorder [34]. Therefore, we are also aiming to create an apnea disorder detection system that uses the VLF of the heart rate interval obtained from BCG. In Section 1, we hypothesized that the head is less affected by body position than the arms, but we have not proved it. In the future, we would like to study the changes in features depending on the body position.

## Figures and Tables

**Figure 1 sensors-21-00952-f001:**
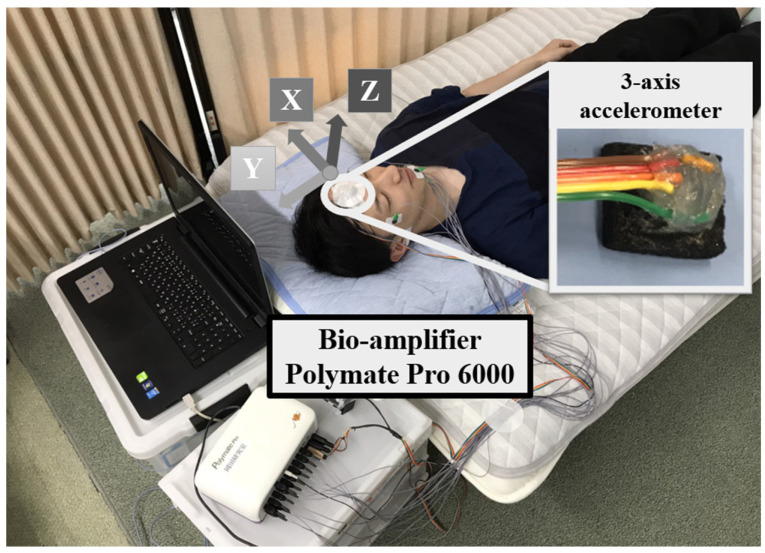
Example of the measurement environment.

**Figure 2 sensors-21-00952-f002:**
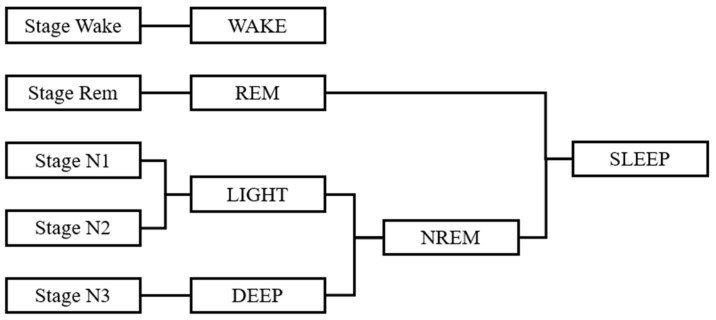
Classification of sleep stages.

**Figure 3 sensors-21-00952-f003:**
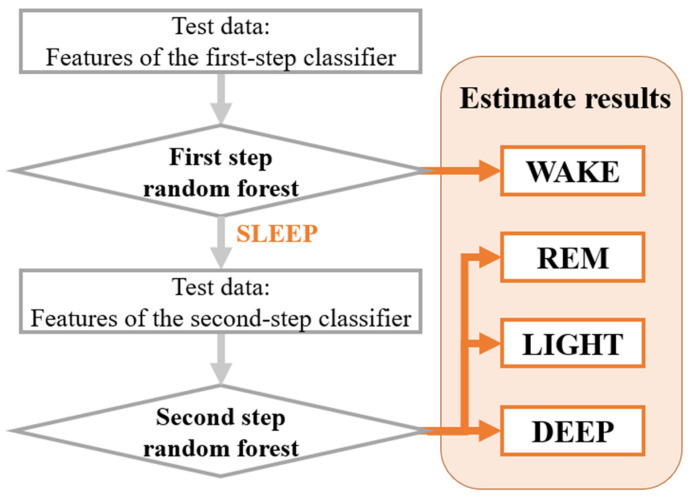
Two-step estimation method.

**Figure 4 sensors-21-00952-f004:**
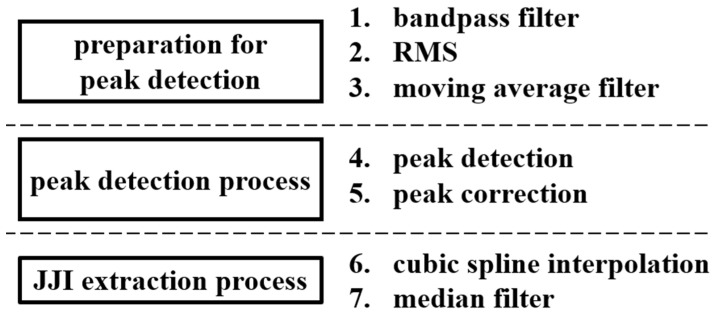
JJI extraction method.

**Figure 5 sensors-21-00952-f005:**
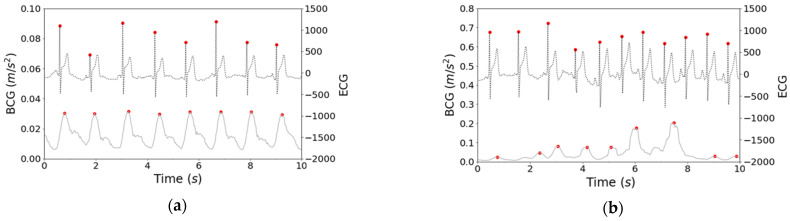
Comparison of BCG and ECG: (**a**) Resting state; (**b**) State including gross movements.

**Figure 6 sensors-21-00952-f006:**
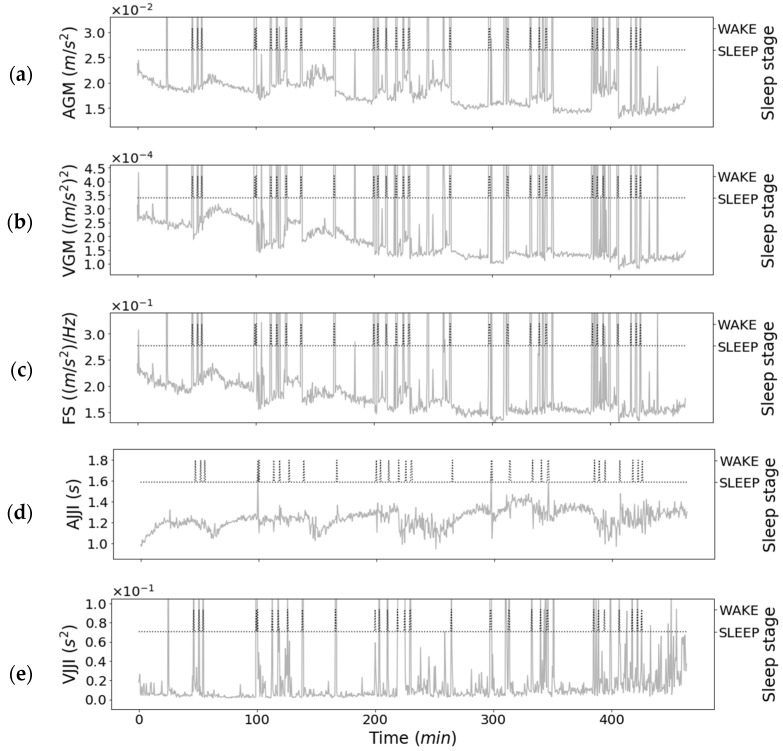
One subject’s results for overnight features with first-step classifier and sleep stage. (Subject: 1, Day: 2): (**a**) Average of gross movements; (**b**) Variance of gross movements; (**c**) Full spectrum; (**d**) Average of JJI; (**e**) Variance of JJI.

**Figure 7 sensors-21-00952-f007:**
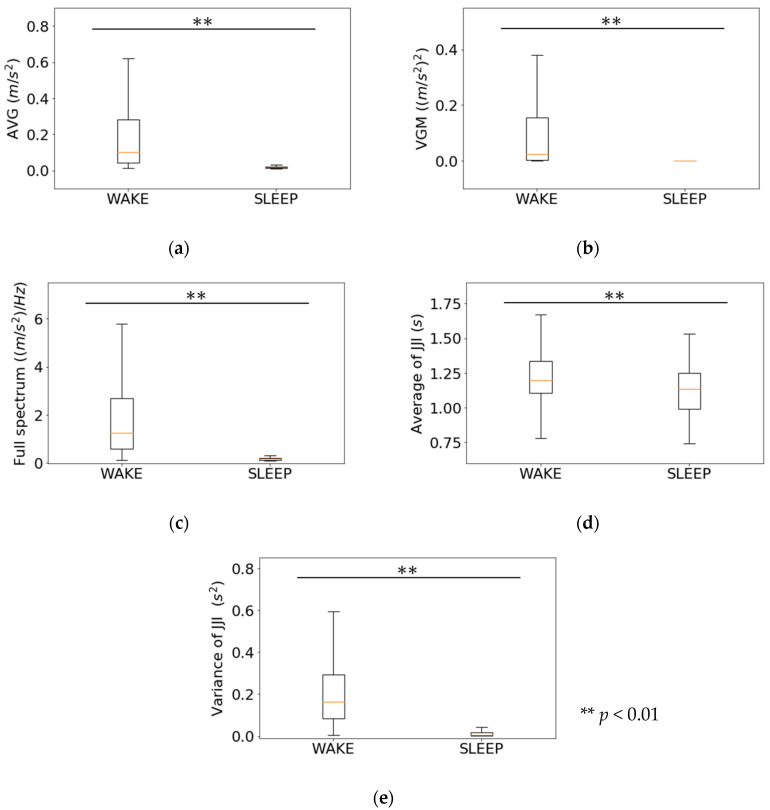
Wilcoxon signed-rank test results for the first-step features: (**a**) Average of gross movements; (**b**) Variance of gross movements; (**c**) Full spectrum; (**d**) Average of JJI; (**e**) Variance of JJI.

**Figure 8 sensors-21-00952-f008:**
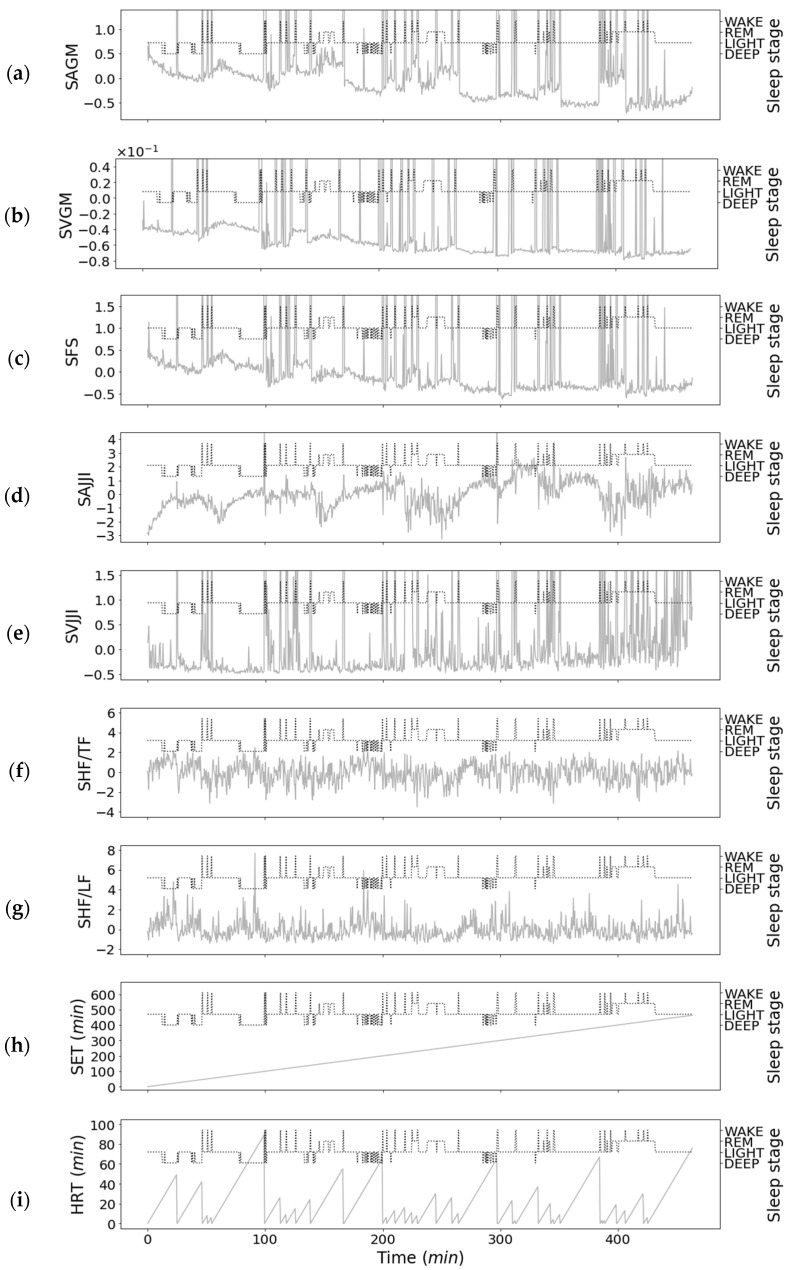
One subject’s overnight results with second-step classifier and sleep stage (Subject: 1, Day: 2): (**a**) Standardized average of gross movements; (**b**) Standardized variance of gross movements; (**c**) Standardized full spectrum; (**d**) Standardized average of JJI; (**e**) Standardized variance of JJI; (**f**) Standardized HF/TF; (**g**) Standardized HF/LF; (**h**) Sleep elapsed time; (**i**) Head rest time.

**Figure 9 sensors-21-00952-f009:**
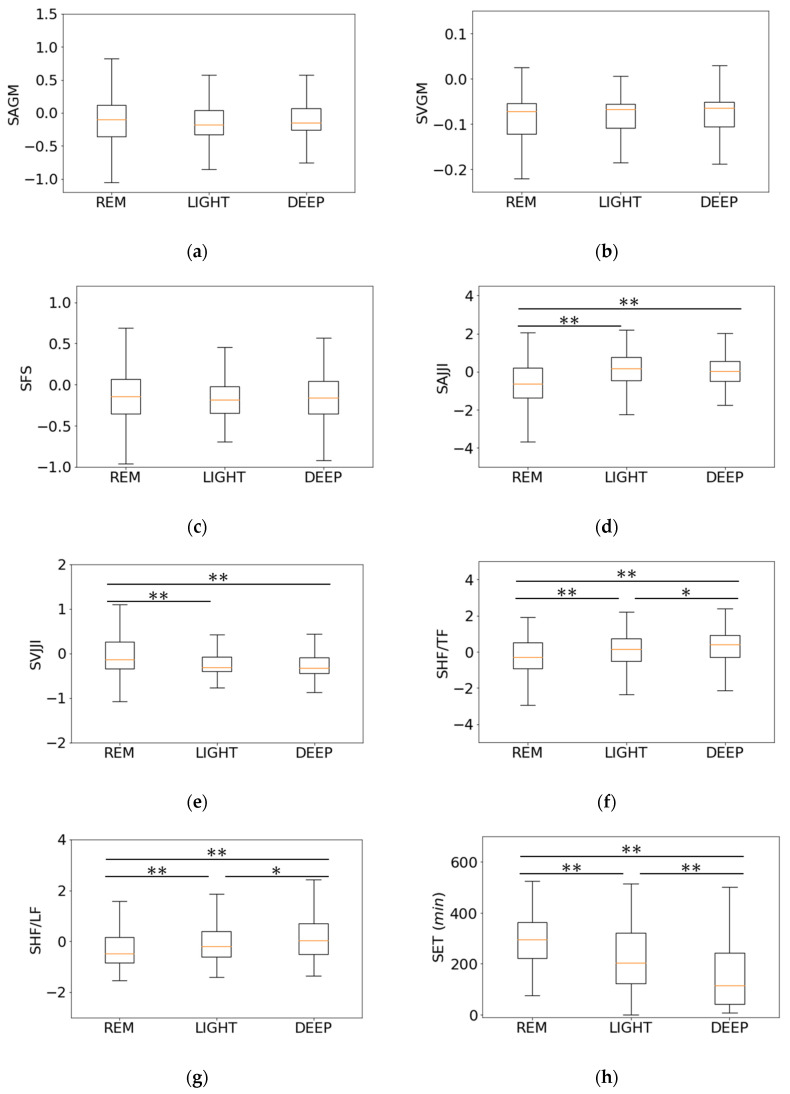
Wilcoxon signed-rank test results for first-step features: (**a**) Standardized average of gross movements; (**b**) Standardized variance of gross movements; (**c**) Standardized full spectrum; (**d**) Standardized average of JJI; (**e**) Standardized variance of JJI; (**f**) Standardized HF/TF; (**g**) Standardized HF/LF; (**h**) Sleep elapsed time; (**i**) Head rest time.

**Figure 10 sensors-21-00952-f010:**
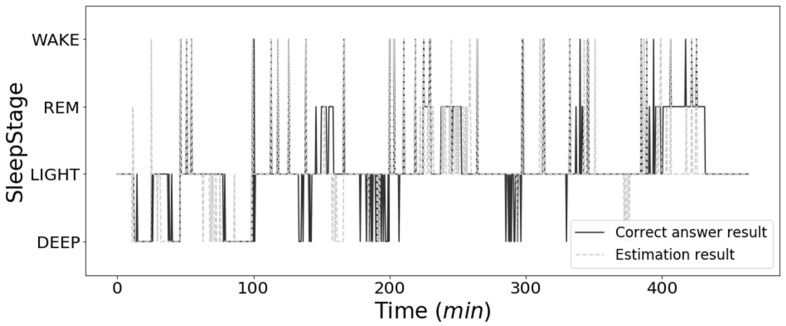
One subject’s overnight estimated results.

**Figure 11 sensors-21-00952-f011:**
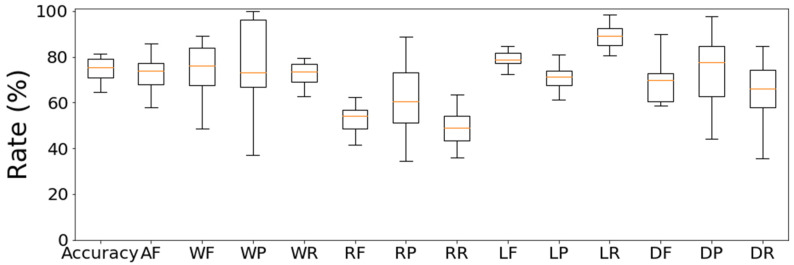
The estimated results of accuracy, F-score, precision, and recall (*n* = 15).

**Figure 12 sensors-21-00952-f012:**
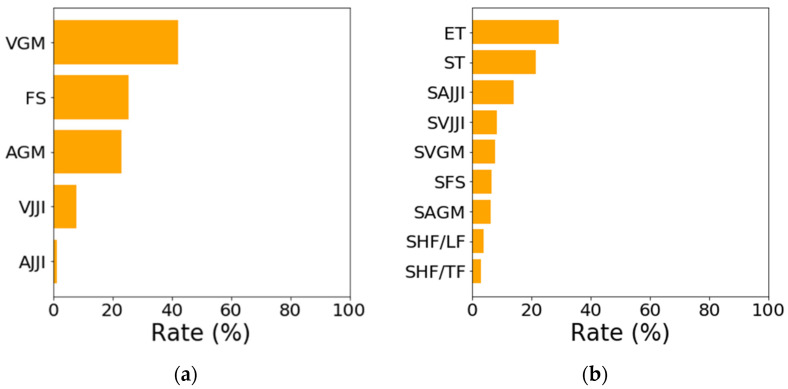
Random forest contribution rates: (**a**) First-step classifier; (**b**) Second-step classifier.

**Figure 13 sensors-21-00952-f013:**
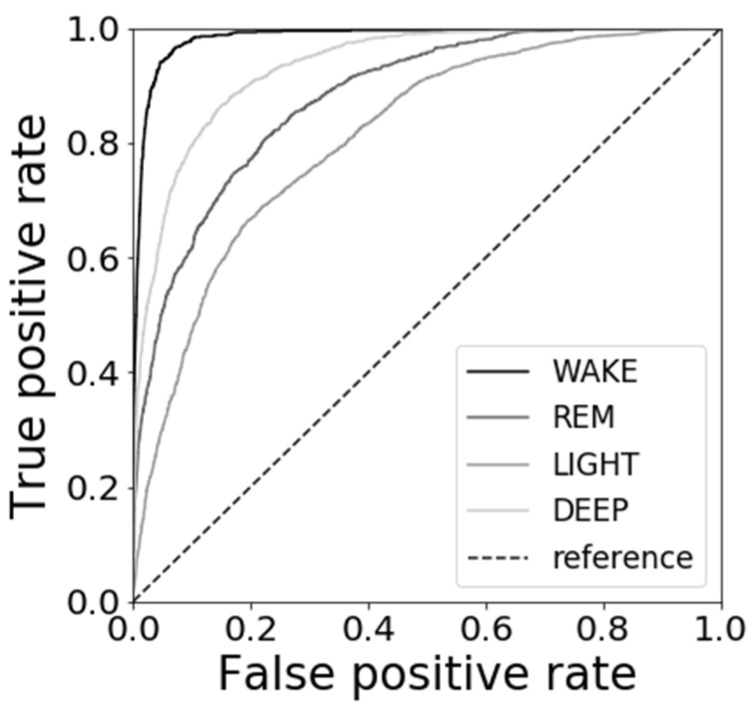
Results of ROC curves.

**Table 1 sensors-21-00952-t001:** Subject information.

Subject No.	1	2	3	4	5	6	7	8	Mean	SD
Age (years)	21	22	22	23	22	21	21	22	21.8	0.7
Height (m)	1.79	1.72	1.70	1.79	1.70	1.65	1.69	1.70	1.718	0.046
Weight (kg)	83	54	68	67	85	55	63	60	66.9	10.9

SD: standard deviation.

**Table 2 sensors-21-00952-t002:** Features of the first-step classifier.

No.	Name	Extraction Method
1	Average of gross movements (AGM)	Gross movements
2	Variance of gross movements (VGM)	Gross movements
3	Full spectrum (FS)	Spectrum
4	Average of JJI (AJJI)	Twitch movements
5	Variance of JJI(VJJI)	Twitch movements

**Table 3 sensors-21-00952-t003:** Features used in the second-step classifier.

No.	Name	Feature Effective for Sleep Stage Estimation
1	Standardized average of gross movements (SAGM)	DEEP
2	Standardized variance of gross movements (SVGM)	DEEP
3	Standardized full spectrum (SFS)	REM, LIGHT DEEP
4	Standardized average of JJI (SAJJI)	REM, LIGHT DEEP
5	Standardized variance of JJI (SVJJI)	REM
6	Standardized HF/TF (SHF/TF)	REM, LIGHT, DEEP
7	Standardized HF/LF (SHF/LF)	REM, LIGHT, DEEP
8	Sleep elapsed time (SET)	REM, LIGHT, DEEP
9	Head rest time (HRT)	DEEP

**Table 4 sensors-21-00952-t004:** Results of sleep stages used as correct data.

Subject No.	1	2	3	4	5	6	7	8	Mean	SD	%Rate
Day 2											
WAKE (epoch)	33	74	95	53	31	52	32	74	55.5	22.1	6.4
REM (epoch)	123	99	85	95	101	135	108	137	110.4	18.0	12.8
LIGHT (epoch)	653	515	471	488	537	597	443	575	534.9	65.7	62.0
DEEP (epoch)	119	206	175	184	141	140	157	170	161.5	26.1	18.7
Day 3											
WAKE (epoch)	32	60	85	65	21	57	38	33	48.9	20.0	5.7
REM (epoch)	121	102	4	63	141	156	161	123	108.9	49.3	12.7
LIGHT (epoch)	597	421	581	472	517	562	528	653	541.4	68.6	63.0
DEEP (epoch)	150	197	98	138	129	279	171	119	160.1	53.3	18.6

SD: standard deviation.

**Table 5 sensors-21-00952-t005:** Confusion matrix obtained from leave-one-out cross-validation (*n* = 15).

		Correct Answer Result			
		WAKE	REM	LIGHT	DEEP
Estimation result	WAKE	604	17	170	27
	REM	11	975	627	47
	LIGHT	131	685	6613	750
	DEEP	4	73	619	1651

**Table 6 sensors-21-00952-t006:** Performance of sleep stage classification in related works for reference.

References (Years)	Sensor (Number)	Modality	*n*	Accuracy (%)	F-score (%)
Mitsukura [35] (2020)	Bed leg BCG sensors (1–4)	BCG	25	89	-
Willemen [36] (2014)	ECG (1), RIP (1), DynaSleep system (1)	Heart rate, Breathing rate, Body movements	85	69	-
Zhang [37] (2018)	Wrist type sensor (1)	Heart rate, Wrist actigraphy	39	-	62
Nochino [38] (2019)	Camera (1)	Body movements	23	41	-
This study	3-axis accelerometer (1)	BCG, Head movements	15	77	73

RIP: Respiratory inductance plethysmography.

## Data Availability

The data presented in this study are available on request from the corresponding author. The data are not publicly available due to privacy or ethical.

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
