# Peer review of "Estimating Sleep Stages Using a Head Acceleration Sensor"

_sensors, 2021, doi:10.3390/s21030952_

Round 1
Reviewer 1 Report
- One of the authors is from Samsung and yet you state at the end of the manuscript that the authors have no conflict of interest to declare. Is there truly no connection?
- You state the company names of the experimental equipment used, but you need to also add the city and country where they are located. For example, labs with the name Digitex can be found in multiple countries. Hence, you should write: Polymate Pro 6000 (Digitex Lab. Co., Ltd., Tokyo, Japan), if that’s the one whose equipment you had used.
- You say that, ‘Sleep is affected by age, gender, and working stress.’ In your study, the age can be considered similar across all 8 subjects. And, they were all male. So that’s good. But how would document that they were under similar levels of stress? Was the study conducted before (or during) an exam period? How many of them were in relationships and how many single? It would be interesting to analyze the results regarding the stress levels they were experiencing at the time. You could have prepared a questionnaire that would assess that. With sample size so low, one must make sure that the subjects are consistent. Perhaps, in the very least, you could discuss that in a discussion section.
- Telling subjects not to drink 3 hours before the experiment is to eliminate their need to urinate during the experiment?
- Did you also tell them not to consume any alcoholic beverage even the day before?
- Did you document how much sleep they had the night before the experiment?
- Were they required to sleep on their backs? Wouldn’t the results be skewed if you include people who normally tend to sleep on their stomach? Have you classified that?
Author Response
Dear Reviewer 1
We really appreciate your careful peer review.
We have taken all your advice into consideration and revised our manuscript, please kindly check the following response. In addition, there is a marker for the corrected part.
Comment #1
One of the authors is from Samsung and yet you state at the end of the manuscript that he authors have no conflict of interest to declare. Is there truly no connection?
Response:
Thank you for pointing out. We lacked our own knowledge about conflicts of interest. We have made the following corrections, so please check.
Page 19, Line 494-498 
Acknowledgments: We would like to thank Editage (www.editage.com) for English language editing. The authors would like to thank Samsung R&D Institute Japan for technical support.
Ethical statement: The study was conducted with the approval of the BKC Research Ethics Review Committee (BKC-2019-076).
Conflicts of Interest: Toshihiro works for Samsung R&D Institute Japan, but there is no conflict of interest.
Comment #2
You state the company names of the experimental equipment used, but you need to also add the city and country where they are located. For example, labs with the name Digitex can be found in multiple countries. Hence, you should write: Polymate Pro 6000 (Digitex Lab. Co., Ltd., Tokyo, Japan), if that’s the one whose equipment you had used.
Response:
I'm sorry for such a mistake. Thank you for checking in detail. It has been modified as follows.
Page 3, Line 110-112 
The experimental equipment used were a biological amplifier Polymate Pro 6000 (Digitex Lab. Co., Ltd., Tokyo, Japan) and the KXM52-1050, 3-axis accelerometer module (Kionix, Tokyo, Japan measurement range ± 19.6 m/s2, sensitivity 0.66 V / (m/s2)).
Comment #3
How would document that they were under similar levels of stress? Was the study conducted before (or during) an exam period? How many of them were in relationships and how many single? It would be interesting to analyze the results regarding the stress levels they were experiencing at the time. You could have prepared a questionnaire that would assess that. With sample size so low, one must make sure that the subjects are consistent. Perhaps, in the very least, you could discuss that in a discussion section.
Response:
Thank you for your advice.
We have added restraint conditions that were not mentioned in the following locations. However, we did not limit drinks 3 hours before the experiment. we think it is necessary in experiments with the elderly, but it is rare for healthy adult men to go to the toilet at night when they do not have problems such as sleep disorders. We asked the subjects in advance if they would go to the toilet frequently at night, but they said that they rarely went to the toilet. For this reason, we assumed that it was not necessary for healthy adult men. As described in the previous studies [6,36], there are some that restrict drinking, naps, and lifestyle habits, but do not restrict drinking 3 hours ago. In fact, even in this experiment, we were instructed to go to the toilet before the measurement, so no one went to the toilet during the measurement.
Page 3, Line 103-105 
Subjects were instructed to go to bed by 23:00 for one week before the experiment. Furthermore, subjects were restricted from caffeine intake, drinking, and excessive exercise for 3 hours before the experiment. In addition, each subject decided the experiment date to reduce their stress.
Page 17, Line 382-383 
As shown in Section 2.2, subjects were given various restrictions to improve their consistency.
Comment #4
Were they required to sleep on their backs? Wouldn’t the results be skewed if you include people who normally tend to sleep on their stomach? Have you classified that?
Response:
Thank you for your comment. In this experiment, we did not limit the position during sleep. The reason is described below.
Page 2, Line 65-76
There are three reasons why we focused on the head among the twitch movements. Firstly, according to Kato et al., the rate of gross movements in overall sleep is 39.39% for the limbs, 27.43% for the trunk, and 22.91% for the head [9]. Therefore, when measuring the pulsation with the arm, it is possible that body movement noise due to the movement of the limbs will be included and the pulsation measurement system will be lowered. Also, since the head has less large movements than the arms, the influence of pulsation measurement is small. . Secondly, a study by Yousefian et al. found that wrist BCG was affected by arm mass, spinal damping, and arm stiffness [12]. In the lateral decubitus position, the trunk may fix the arm and make BCG measurement difficult. The movement of the head by BCG is a vertical movement [13], and it is not easily affected by the posture during sleep. Therefore, the measurement of BCG during sleep is better on the head than on the wrist. Finally, the twitch movements of the head can be measured by using an earphone-type accelerometer[8] or camera [13,14], which is practical.
Page 19, Line 486-488
In Section 1, we hypothesized that the head is less affected by body position than the arms, but we have not proved it. In the future, we would like to study the changes in features depending on the body position.
Reviewer 2 Report
Research reports estimating sleep stages using a head acceleration sensor. The introduction need to be further improved with the latest literature review.
1) Evaluation and methods used for classification need to be verified
2) Authors mention Kfold cross validation is not feasible due to two stage classification method. But from the results it is not clear how the whole classification method has done
3) ROC analysis including sensitivity and specificity need to be included in the revised version
4) Compare your method with the other state of the art methods available in the literature.
5) Conclusion section need to be further improved.
Reviewer 3 Report
The paper presents an interesting subject, but the following improvements must be addressed in order to increase the scientific soundness of the paper:
- there is no section with related work
- it is necessary to explain why is necessary to use another sensor to detect sleep stages. There are already a set of devices (Fitbit, Emfit QS sensor, etc) that are easier to use and they detect sleep stages for different types of users
- the experimental phase must include explanation about validation phase of sleep
- comparison with other existing sensors must be added
Round 2
Reviewer 1 Report
Thank you for addressing my previous comments. I have no additional feedback.
Author Response
Dear Reviewer 1
Thank you for your careful peer review. Thank you for making the paper better thanks to Reviewer 1.
We have made minor amendments to this paper. In addition, you can see the change history in "Track Changes" function in Microsoft Word.
Reviewer 2 Report
The authors have addressed all my comments satisfactorily and the paper can be considered for publication.
Author Response
Dear Reviewer 2
Thank you for your careful peer review. Thank you for making the paper better thanks to Reviewer 2.
We have made minor amendments to this paper. In addition, you can see the change history in "Track Changes" function in Microsoft Word.
Reviewer 3 Report
My comments were addressed clearly and detailed. I recommend to publish the paper.
Author Response
Dear Reviewer 3
Thank you for your careful peer review. Thank you for making the paper better thanks to Reviewer 3.
We have made minor amendments to this paper. In addition, you can see the change history in "Track Changes" function in Microsoft Word.